# The TGF-β/SMAD Signaling Pathway Prevents Follicular Atresia by Upregulating MORC2

**DOI:** 10.3390/ijms231810657

**Published:** 2022-09-13

**Authors:** Jiying Liu, Nannan Qi, Wenwen Xing, Mengxuan Li, Yonghang Qian, Gang Luo, Shali Yu

**Affiliations:** 1School of Biotechnology, Jiangsu University of Science and Technology, Zhenjiang 212018, China; 2Department of Occupational Medicine and Environmental Toxicology, Nantong Key Laboratory of Environmental Toxicology, School of Public Health, Nantong University, Nantong 226019, China

**Keywords:** porcine, follicular atresia, MORC2, TGF-β/SMAD signaling pathway

## Abstract

In mammals, female fertility is determined by the outcome of follicular development (ovulation or atresia). The TGF-β/SMAD signaling pathway is an important regulator of this outcome. However, the molecular mechanism by which the TGF-β/SMAD signaling pathway regulates porcine follicular atresia has not been fully elucidated. Microrchidia family CW-type zinc finger 2 (MORC2) is anovel epigenetic regulatory protein widely expressed in plants, nematodes, and mammals. Our previous studies showed that *MORC2* is a potential downstream target gene of the TGF-β/SMAD signaling pathway. However, the role of *MORC2* in porcine follicular atresia is unknown. To investigate this, qRT-PCR, western blotting, and TdT-mediated dUTP nick-end labeling were performed. Additionally, the luciferase activity assay was conductedto confirm that the TGF-β/SMAD signaling pathway regulates MORC2. Our results demonstrate that MORC2 is animportant anti-apoptotic molecule that prevents porcine follicular atresia via a pathway involving mitochondrial apoptosis, not DNA repair. Notably, this studyrevealsthat the TGF-β/SMAD signaling pathway inhibits porcine granulosa cell apoptosis by up-regulating MORC2. The transcription factor SMAD4 regulated the expression of *MORC2* by binding to its promoter. Our results will help to reveal the mechanism underlying porcine follicular atresia and improve the reproductive efficiency of sows.

## 1. Introduction

The reproductive traits of sows seriously affect the economic benefits of the pig industry. Ovarian follicular development and ovulation are the main factors that affect female reproductive performance. Fewer than 1% of follicles undergo ovulation, and the remainder undergoes atresia and degradation, leading to a huge waste of reproductive resources. The reproductive potential of female livestock could be enhanced if follicular atresia could be prevented. The causes of follicular atresia are complex. Most research has focused on cell apoptosis, autophagy, oxidative stress, hypoxia, non-coding RNA, and other aspects [1,2,3,4,5]. However, the essential factors that cause follicular atresia remain unclear. Follicular atresia limits the fertility of sows, and granulosa cell apoptosis is a hallmark of this process. The TGF-β/SMAD signaling pathway is closely related to female fertility. Core factors involved in this pathway, such as TGF-β1, TGFBR2, SMAD2/3, SMAD1/5, and SMAD4, are key candidates to affect female reproduction [6,7,8,9,10,11]. Moreover, the TGF-β/SMAD signaling pathway affects follicular atresia in pigs [11,12,13]. Activation of the TGF-β/SMAD signaling pathway inhibits apoptosis of follicular granulosa cells, while blockade of this pathway has the opposite effect [14,15]. However, the molecular mechanism by which the TGF-β/SMAD signaling pathway regulates granulosa cell apoptosis has not been fully elucidated. 

Microrchidia (MORC) family proteins, which were first reported and characterized in germ cells, are members of the ATPase family and are critical for life. Research into the functions of MORC proteins is just beginning. MORCs, including MORC1, MORC2, MORC3, and MORC4, are linked to immunological disorders, neurogenic disorders, and cancer [16,17,18,19]. In addition, recent research showed that MORC3 has a primary anti-viral function [17,19]. Generally speaking, MORCs contain a conserved GHKL-ATPase domain at the N-terminus, a conserved CW-type zinc finger domain in the middle, and several coiled-coil domains. At present, people pay more attention to MORC2 and MORC3 proteins. MORC2, also known as ZCWCC1, is widely expressed, especially in the ovary, testis, brain, uterus, and stomach [20]. As an epigenetic regulatory protein, the function of MORC2 has not been fully elucidated but is usually associated with gene silencing, chromatin remodeling, and DNA repair, and it affects cell survival and participates in the regulation of cell apoptosis [21,22,23,24,25]. In addition, compared with other MORC proteins, MORC2 has unique structural characteristics. The nuclear localization signal and proline-rich domain of human MORC2 play a role in transcriptional inhibition in gastric and colon cancer cells. However, research on MORC2 is in its infancy. Mutation of *MORC2* is associated with Charcot-Marie-tooth disease and neuro-developmental disorders [16]. In addition, MORC2 is highly expressed in many cancers [20]. The function of MORC2 in normal somatic and germ cells remains poorly understood. In 2019, Liu et al. found that MORC2 inhibits differentiation and promotes the proliferation of C2C12 mouse skeletal muscle cells [26]. Moreover, MORC2 is highly expressed in mouse ovaries and testes, and the inactivation of its homologous gene (*MORC2b*) blocks ovarian development and prevents the formation of mature follicles [27]. This suggests that MORC2 plays an important role in the regulation of mammalian ovarian development and reproduction. Our previous studies showed that TGF-β1 inhibits apoptosis of porcine follicular granulosa cells [11]. Meantime, transcriptomic studies identified 1025 differentially expressed genes in porcine follicular granulosa cells upon SMAD4 interference, of which 530 were up-regulated and 495 were down-regulated [28]. Among which, *MORC2* was significantly down-regulated, indicating there is an association between the TGF-β/SMAD signaling pathway and MORC2, which needs to be studied further. While there is no report about the function of *MORC2* in large domestic animals such as pigs.

This study aimed to explore the effect of MORC2 on porcine granulosa cell apoptosis and the role in the process of porcine follicular atresia regulated by the TGF-β/SMAD4 signaling pathway. In this study, we discovered that MORC2 is an anti-apoptotic factor inporcine granulosa cells. Functional assays demonstrated that MORC2 inhibits cell apoptosis via the mitochondrial apoptosis pathway, not the DNA repair pathway. Moreover, we revealed that the TGF-β/SMAD signaling pathway up-regulates MORC2. The current research intended to explore the effect of the *MORC2* gene on porcine granulosa cell apoptosis and elaborated on the physiological role of the TGF-β/SMAD4 signaling pathway in porcine follicular atresia.

## 2. Results

### 2.1. MORC2 Inhibits Apoptosis of Porcine Follicular Granulosa Cells

MORC2 plays an important role in the regulation of cell fate. We aimed to knock down or over-express MORC2in order to elucidate its function in the apoptosis of porcine follicular granulosa cells. An effective MORC2-targeting siRNA (MORC2-siRNA) was screened by qRT-PCR (Figure 1A) and western blotting (WB, Figure 1B). That indicated MORC2-siRNA-1164 wasthe effective siRNA. qRT-PCR showed that MORC2 knockdown significantly reduced the *Bcl2/Bax* ratio (Figure 1C), but did not affect the transcription levels of *Caspase-3* and *PARP-1* (Figure 1D). WB showed that cleavage of Caspase-3 and PARP-1 was promoted by MORC2 knockdown but inhibited by MORC2 over-expression (Figure 1E,F). In addition, TdT-mediated dUTP nick-end labeling (TUNEL) staining showed that MORC2 knockdown significantly promoted apoptosis of porcine granulosa cells (Figure 2A,B). These results suggest that MORC2 is an anti-apoptotic protein in granulosa cells.

### 2.2. MORC2 Inhibits Apoptosis of Porcine Granulosa Cells Independently of the DNA Repair Pathway

The DNA damage response is essential for the proper functions of all cells and organisms. MORC2, which contains an ATPase module, has been mechanistically linked to the DNA damage response. To explore the function of MORC2, we examined whether it affects the DNA damage response. Ataxia-telangiectasia mutated (ATM) protein kinase, Ataxia-telangiectasia and Rad3-related (ATR), p53, and Rad51 are the main factors involved in the DNA damage response. qRT-PCR showed that knockdown of MORC2 did not significantly alter the expression of these factors (Figure 3). These results indicate that MORC2 inhibits granulosa cell apoptosis independently of the DNA repair pathway.

### 2.3. MORC2 Is a Downstream Target of the TGF-β/SMAD Signaling Pathway

Our previous results demonstrated that the TGF-β/SMAD signaling pathway is critical for the determination of ovarian fate and SMAD4 is a core component of this pathway. Based on RNA-seq data (GSE65696) obtained after the knockdown of SMAD4 in porcine granulosa cells [28], we found that SMAD4 knockdown significantly down-regulated MORC2 (Figure 4A). This was confirmed by qRT-PCR (Figure 4B) and WB (Figure 4C). On the contrary, SMAD4 over-expression dramatically increased the MORC2 protein level (Figure 4D). Similarly, TGF-β1 treatment increased MORC2 protein expression (Figure 4E). These results suggest that the TGF-β/SMAD pathway induces MORC2 expression in porcine granulosa cells. So, does TGF-β/SMAD signaling pathway regulates porcine granulosa cell apoptosis through MORC2? The expression levels of Caspase-3 and PARP-1 were detected after knockdown and over-expression of SMAD4. The results showed that the knockdown of SMAD4 promoted the cleavage of Caspase-3 and PARP-1 (Figure 5A), but over-expression of SMAD4 had the opposite effect (Figure 5B), which further supported our hypothesis.

### 2.4. The Transcription Factor SMAD4 Directly Binds to the MORC2Promoter

SMAD4 is an important transcription factor that regulates the expression of many genes. We previously reported that the core promoter region of the porcine *MORC2* gene is located between -1207 and -1347 in the 5′ regulatory region (ATG is +1) [29]. To study the effect of the TGF-β/SMAD signaling pathway on MORC2 expression, smad binding elements (SBE) sites were predicted using PROMO and JASPAR online software. Two SBE sites in the promoter of *MORC2* were identified (Figure 6A). SBE1 was contained in the pGL3-1479 vector and SBE2 was contained in the pGL3-1347 vector. SMAD4 over-expression increased relative luciferase activity of pGL3-1479, but not of pGL3-1347 (Figure 6B). This indicates that SMAD4 binds to SBE1. To verify this, the binding sites were deleted (Figure 6C). SMAD4 over-expression did not affect the luciferase activity of the MORC2 mutant promoter (Figure 6D). Relative luciferase activity of pGL3-1479 was significantly increased by treatment with 10 and 20 ng/mL TGF-β1, but not by treatment with 5 ng/mL TGF-β1 (Figure 6E), whereas relative luciferase activity of the MORC2 mutant promoter was not affected (Figure 6F). These results indicate that the transcription factor SMAD4 binds to SBE1 of the *MORC2* promoter.

## 3. Discussion

In 1996, the first member of the MORC family was identified in testes and named MORC1. MORC1 is mainly expressed in embryonic stem cells and thymocytes, MORC2 and MORC3 are expressed ubiquitously, and MORC4 is mainly expressed in the placenta and pituitary gland. Decades of research have revealed that not only MORC1, but also MORC2 and MORC3 are closely related to the regulation of animal reproduction [27,30,31]. However, most studies of MORC proteins mainly concentrated on male reproduction. MORC1 is specifically expressed in the male germline and its ablation results in male sterility with meiotic arrest [32,33]. In addition, loss of MORC3 reduces the pregnancy rate and leads to subfertility in male mice but does not affect the litter size of female mice [34]. However, *MORC2* is not only related to male reproduction but also to female reproduction. Both male and female *Morc2b*^−/−^ mice are sterile. Ovaries of adult *Morc2b*^−/−^ female mice are much smaller than those of heterozygous littermates and lack oocytes [27]. This indicates that MORC2 is important for female reproduction. MORC1 induces cell apoptosis in mouse testes [31] and MORC2 mediates cell apoptosis in neurons and oocytes [16,27]. However, it is unknown whether MORC2 affects the apoptosis of porcine granulosa cells. Here, we reported that MORC2 is an anti-apoptotic protein in porcine granulosa cells. Apoptosis of granulosa cells increased when *MORC2* was knocked down. This indicates that MORC2 is an anti-apoptotic factor in porcine granulosa cells. The DNA damage response is a process by which DNA is repaired via the actions of various enzymes. ATM, ART, p53, and Rad51 are markers of DNA repair. *MORC2* was reported to be a DNA damage response gene [16,22,35]. Therefore, we investigated whether MORC2 inhibits porcine granulosa cell apoptosis via the DNA repair response. MORC2 knockdown did not affect the expression of DNA repair response genes. Meanwhile, knockdown of MORC2 significantly increased expression of the pro-apoptotic gene *Bax* as well as the levels of cleaved Caspase-3 and PARP-1, indicating that MORC2 inhibits apoptosis via the mitochondrial apoptosis pathway. MORC2 may be a candidate marker of reproductive performance because it decreases porcine granulosa cell apoptosis.

Atresia occurs in follicles at all stages. Many studies have revealed the factors that cause follicular atresia. However, these are just the tip of the iceberg, and many unknown factors await further investigation. In porcine ovaries, the ovulation rate is an important parameter for the reproductive efficiency of sows. In vivo and in vitro experiments revealed that gonadal steroids, growth factors, and cytokines are essential for follicular development. Among various intra-ovarian paracrine and autocrine factors, members of the TGF-β superfamily are best studied in terms of their effects on follicular atresia. TGF-β1 is an important member of this superfamily and is crucial for granulosa determination of granulose cell fate and female fertility [14,36,37]. The *TGF-β1* gene is significantly associated with the reproductive performance of Large White sows [37] and is involved in porcine granulosa cell apoptosis [14].The results of TGF-β1 regulating granulosa cell apoptosis are inconsistent. In porcine ovarian granulosa cells, TGF-β1 inhibits granulosa cell apoptosis [14]. However, TGF-β1 promotes granulosa cell apoptosis in polycystic ovary syndrome [38] and bovine granulosa cells [39]. Therefore, TGF-β1 not only inhibits but also promotes granulosa cell apoptosis. This indicates that the functions of TGF-β1 differ according to the physiological conditions. SMAD4 is the only common mediator of Smad (Co-SMAD), and accumulating evidence indicates that it is associated with ovarian development, especially granulosa cell apoptosis [11,40,41]. In general, SMAD4 and regulatory SMADs (R-SMADs) form transcriptional complexes and regulate gene expression by binding to promoters of genes in the nucleus. Here, we identified a binding site for SMAD4 in the *MORC2* promoter; therefore, *MORC2* is a novel gene regulated by the TGF-β signaling pathway.

Research on MORC2 has mainly focused on its function and structure. MORC2 mutations cause Charcot-Marie-Tooth neuropathy type 2Z disease [42], and recent research showed that MORC2 is aberrantly highly expressed in many cancers, including breast cancer [25,35], hepatocellular carcinoma cells [18], and gastric cancer cells [43]. Although the present study revealed the role of MORC2, further studies are needed. The function of MORC2 in porcine granulose cell apoptosis of large domestic animals (pigs) was preliminarily investigated in this study, and the effect of MORC2 on the reproductive performance of pigs will be further explored. MORC2 contains a highly conserved GHKL-ATPase domain, a CC domain, and a CW zinc finger domain. Thus, MORC2 may function in transcription regulation, chromatin remodeling, and DNA repair. Therefore, it is necessary to study the regulation of MORC2 expression. However, research has mainly focused on its regulation at the post-transcriptional level. For instance, GPER1 phosphorylates threonine 582 of MORC2 [23]. MORC2 767 lysine could be acetylated by the acetyltransferase NAT10 and deacetylase SIRT2 [35]. In addition, threonine 556 of MORC2 is O-GlcNAcylated by TGF-β1 [25]. However, there are few studies of the upstream regulatory signals of MORC2 expression. For glucose-induced MORC2 expression, c-Myc binds to the promoter of *MORC2* and activates its transcription [44]. Although the promoter of *MORC2* contains two ESR1-binding sites, E2 does not affect its mRNA expression [23]. To reveal the function of MORC2, it is necessary to study its transcriptional regulation. Although a previous study showed that TGF-β1 induces O-GlcNAcylation of MORC2 by regulating GFAT, it was unclear whether TGF-β1 directly regulates MORC2 expression. One of the highlights of this study is that it shows that MORC2 is a direct target of the TGF-β/SMAD signaling pathway.

In summary, the present study provides evidence that MORC2 is an anti-apoptotic factor in porcine granulosa cells. It also shows that the transcription factor SMAD4 binds to the *MORC2* promoter. The findings presented here reveal a previously unrecognized functional and mechanistic role of the TGF-β/SMAD signaling pathway in follicular atresia. Based on our results, we propose a molecular mechanism (Figure 7).

## 4. Materials and Methods

### 4.1. Reagents

DMEM/F12 and penicillin/streptomycin (Life Technologies Co., Carlsbad, CA, USA), HighGene Transfection reagent (ABclonal, Wuhan, China), RNAiso Plus, and a PrimeScript™ RT reagent Kit with gDNA Eraser (Perfect Real Time) (Taraka, Dalian, China), a One Step TUNEL Apoptosis Assay Kit and a BCA Protein Assay Kit (Beyotime Biotechnology, Shanghai, China), a Dual-Luciferase Reporter Assay Kit (Vazyme, Nanjing, China), TGF-β1 (NovoProtein, Suzhou, China), an anti-MORC2 rabbit polyclonal antibody (Sangon Biotech, Shanghai, China), an anti-SMAD4 rabbit polyclonal antibody (Santa Cruz Biotechnology, Shanghai, China), and PVDF membrane (Merck Millipore, Darmstadt, Germany) were purchased from the indicated suppliers.Primers were synthesized by Shangya (Hangzhou, China). siRNAs were synthesized by GenePharma (Shanghai, China). NovoStart^®^ SYBR qPCR SuperMix Plus was purchased from Novoprotein (Nanjing, China).

### 4.2. Isolation and Culture of Porcine Granulosa Cells 

The animal protocols were approved by the Institutional Animal Care and Use Committee (IACUC) of Jiangsu University of Science and Technology (G2022SJ12, Zhenjiang, China). Animal care and handling followed the IACUC guidelines.

Fresh ovaries were collected from 6-month-old commercial sows from the local slaughterhouse. A total of 72 sows were used. Healthy follicles of 3~5 mm were selected to collect the porcine ovarian granulosa cell. The follicles were immediately immersed in saline containing penicillin/streptomycin and transported to the laboratory within 2 h. Porcine granulosa cells were collected from healthy follicles (3~5 mm diameter) using 22-gauge needles. The cells were washed twice with phosphate-buffered saline (PBS) and centrifugation at 800× *g* for 5 min. Then the cells were cultured in DMEM/F12 containing 10% fetal bovine serum, and 1% penicillin-streptomycin at 37 °C in an incubator containing 5% CO_2_.

### 4.3. Plasmid Construction 

The *MORC2* promoter was cloned into the pGL3-basic vector between *Hind*III and *Kpn*I. The SEB sites were predicted using PROMO (http://alggen.lsi.upc.es/cgi-bin/promo_v3/promo/promoinit.cgi?dirDB=TF_8.3 (accessed on 15 June 2021)) and JASPAR (https://jaspar.genereg.net (accessed on 2 July 2021)). The SEB-mut vector was generated using a Mut Express^®^ II Fast Mutagenesis Kit V2 (Vazyme, Nangjing, China) and verified by Sanger sequencing. The coding domain sequence (CDS) region of *MORC2* was inserted into the pcDNA3.1-His-v5 over-expression vector. The CDS of *SMAD4* was inserted into the pcDNA3.1 over-expression vector. The primers used for plasmid construction are listed in Table 1.

### 4.4. Cell Transfection

Porcine granulosa cells were seeded into 12-well plates and transfected when their density reached 70%. The siRNAs are listed in Table 1. Briefly, 50 pmol siRNA and 3 μL High Gene Transfection reagent were mixed in 100 μL serum-free DMEM/F12 and added to the cell culture. The plate was gently shaken. After 4–6 h of transfection, half the medium was replaced by a fresh complete medium. At 24 or 48 h after transfection, cells were analyzed by various methods, such as RT-PCR, WB, and luciferase assays.

### 4.5. Total RNA Extraction and Quantitative PCR 

Total RNA was extracted from porcine granulosa cells using RNAiso Plus. In total, 1 µg total RNA was reverse-transcribed using a PrimeScript™ RT reagent Kit with gDNAEraser according to the manufacturer’s instructions. qRT-PCR was performed using the following conditions: 95 °C for 1 min, followed by 35 cycles of 95 °C for 20 s, 60 °C for 20 s, and 72 °C for 30 s. The primers are listed in Table 2. The relative expression of each gene was calculated using the 2^−ΔΔCt^ method. *GAPDH* was used as a reference gene.

### 4.6. WB Analysis

Whole cells were lysed on ice for 30 min with RIPA buffer containing protease inhibitors. The lysate was centrifuged for 10 min at 4 °C at 12,000× *g*. The total protein concentration was determined using a BCA Protein Assay Kit. In total, 10 µg total protein was loaded per lane, separated by 4–20% SDS-PAGE, and transferred to a PVDF membrane. The membrane was blocked with 5% non-fat milk for 2 hat room temperature, incubated with primary antibodies overnight at 4 °C, washed thrice with TBST, incubated with secondary HRP-conjugated antibodies, washed thrice with TBST, and incubated with ECL reagent. WB images were captured using a chemiluminescence imaging system (Qinxiang, Shanghai, China). β-Tubulin was used as an internal control.

### 4.7. Tunel

Porcine granulosa cells were washed once with phosphate-buffered saline (PBS), fixed with 4% paraformaldehyde for 30 min, washed with PBS, incubated with PBS containing 0.3% Triton X-100 at room temperature for 5 min, washed twice with PBS, incubated with TUNEL solution for 60 min at 37 °C, and washed twice with PBS. Slides were sealed with anti-fluorescence quenching DAPI solution. Cy3 was detected using an excitation wavelength of 550 nm and an emission wavelength of 570 nm (red fluorescence).

### 4.8. Luciferase Reporter Gene Assay

Porcine granulosa cells were collected after transfection for 24 h, lysed using 1× cell lysis buffer, and centrifuged for 10 min at 12,000× *g*. Analysis was performed using a Dual-luciferase Reporter Assay Kit (Vazyme, Nanjing, China). Firefly and Renilla luciferase activities were evaluated and normalized for each sample.

### 4.9. Statistical Analysis 

GraphPad Prism 6 (GraphPad Software, San Diego, CA, USA) was used to perform statistical analysis. The two-tailed Student’s *t*-test was used to compare two groups. All data are presented as the mean ± SEM of at least three independent experiments. *p* < 0.05 was considered statistically significant (*) and *p* < 0.01 was considered highly statistically significant (**).

## 5. Conclusions

We demonstrated that MORC2 regulates porcine granulosa cell apoptosis. The TGF-β/SMAD signaling pathway inhibits porcine granulosa cell apoptosis via binding of SMAD4 to the promoter of *MORC2* and activation of its expression. Our findings provide insight into the mechanism by which the TGF-β/SMAD signaling pathway functions in reproductive regulation. The results provide new ideas and a theoretical basis for reducing follicular atresia and improving the reproductive efficiency of domestic animals and reproductive health.

## Figures and Tables

**Figure 1 ijms-23-10657-f001:**
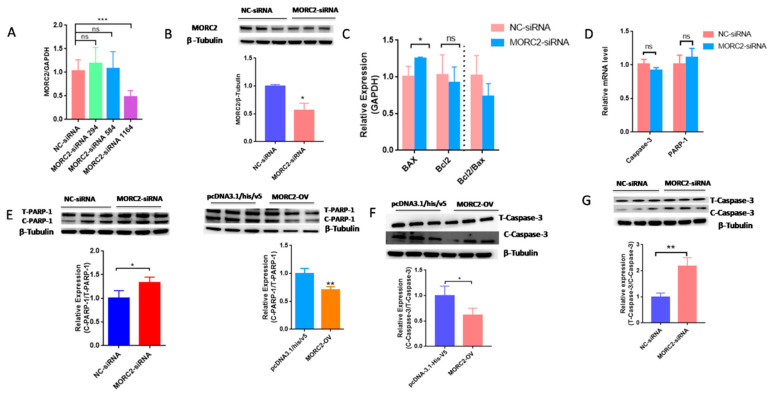
Knockdown of MORC2 induces apoptosis-related gene expression in porcine granulosa cells. (**A**) qRT-PCR analyse of MORC2 mRNA relative expression in porcine granulosa cells transfected with different MORC2-targeting siRNAs. (**B**) The relative protein expression of MORC2 in porcine granulosa cell transfected with MORC2-siRNA. (**C**) qRT-PCR analysis of *Bax*, *Bcl2*, and *Bax/Bcl2* upon depletion of *MORC2* by siRNA. (**D**,**E**) qRT-PCR analysis of *Caspase-3* and *P**ARP-1* in porcine granulosa cells transfected with MORC2-siRNA. *GAPDH* was used as a loading control. (**F**,**G**) Representative WB and quantification of the cleaved Caspase-3 and cleaved PARP-1 levels in porcine granulosa cells depleted of MORC2.NC-siRNA indicates small interfering RNA of the negative control. MORC2-siRNA means small interfering RNA of *MORC2* gene. MORC2-siRNA 294, 584, and 1164 indicate different small interfering RNA target different sites in the *MORC2* gene. MORC2-OV indicates the *MORC2* overexpression vector. T-PARP-1: total protein of PARP-1; C-PARP-1 cleaved protein of PARP-1. T-Caspase-3: total protein of Caspase-3; C-Caspase-3: cleaved protein of Caspase-3. Results are shown as mean ± SEM (n = 3). ns means no significance (*p* > 0.05); * *p* < 0.05, ** *p* < 0.01, *** *p* < 0.001.

**Figure 2 ijms-23-10657-f002:**
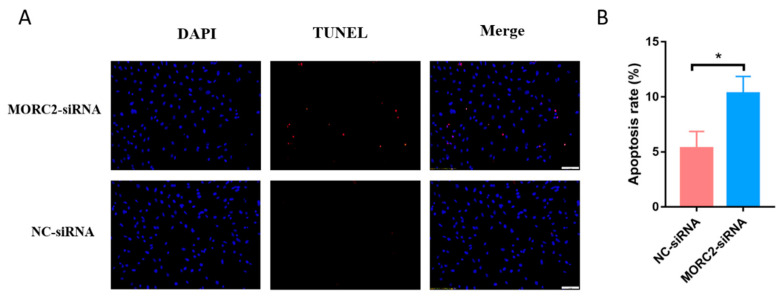
MORC2 attenuates porcine granulosa cell apoptosis. (**A**) TUNEL assay to detect apoptotic porcine granulosa cells transfected with MORC2 siRNA. The Scale bar = 50 μm. (**B**) Quantification of apoptotic porcine granulosa cells upon treatment with MORC2-siRNA. DAPI was used to label nuclei and TUNEL was used to label apoptotic cells. * *p* < 0.05.

**Figure 3 ijms-23-10657-f003:**
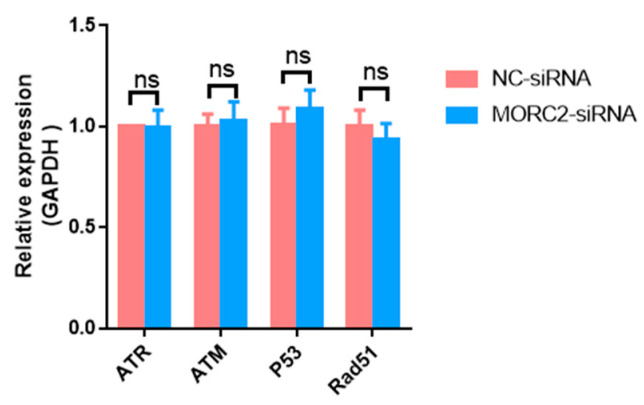
Expression of genes related to the DNA repair pathway. qRT-PCR analysis of mRNA expression of *ATR*, *ATM*, *p53*, and *Rad51* upon knockdown of *MORC2*. *GAPDH* was used as a loading control. ns means no significance (*p* > 0.05).

**Figure 4 ijms-23-10657-f004:**
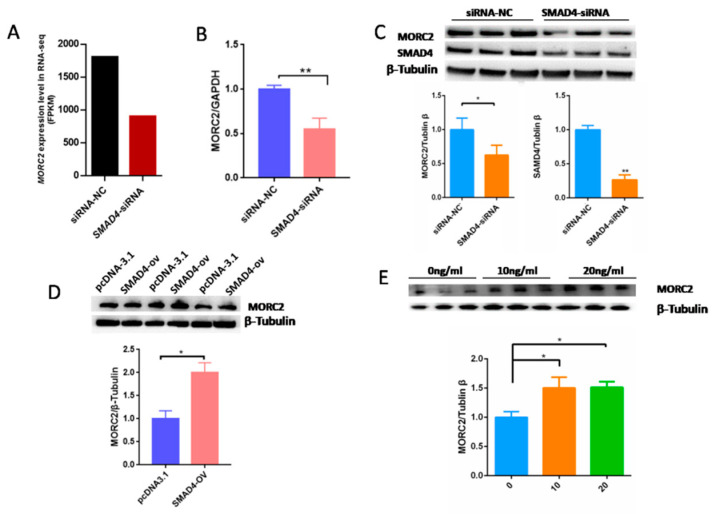
The TGF-β/SMAD signaling pathway significantly up-regulates MORC2 in porcine granulosa cells. (**A**) RNA-seq analysis of the *MORC2* mRNA expression level (fragments per kb of exon per million fragments mapped; FPKM) in porcine granulosa cells treated with negative control siRNA (NC-siRNA) or SMAD4 knockdown RNA (SMAD4-siRNA). (**B**) Confirmation of the RNA-seq result by qRT-PCR after treatment with SMAD4 siRNA. *GAPDH* was used as a loading control. (**C**,**D**) WB analysis of theMORC2 protein level in porcine granulosa cells upon knockdown or over-expression of SMAD4 (SMAD4-OV). (**E**) The MORC2 protein level in porcine granulosa cells treated with TGF-β1. β-Tubulin served as a loading control. Results are shown as mean ± SEM (n = 3). * *p* < 0.05, ** *p* < 0.01.

**Figure 5 ijms-23-10657-f005:**
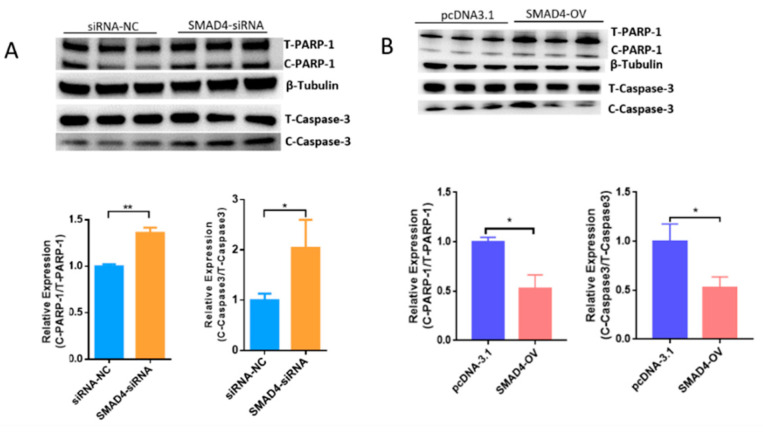
WB analysis of the Caspase-3 and PARP-1 protein level in porcine granulosa cellsupon knockdown (**A**) or over-expression of SMAD4 (**B**). SMAD4-OV: SMAD4 overexpression vector. Results are shown as mean ± SEM (n = 3). * *p* < 0.05, ** *p* < 0.01.

**Figure 6 ijms-23-10657-f006:**
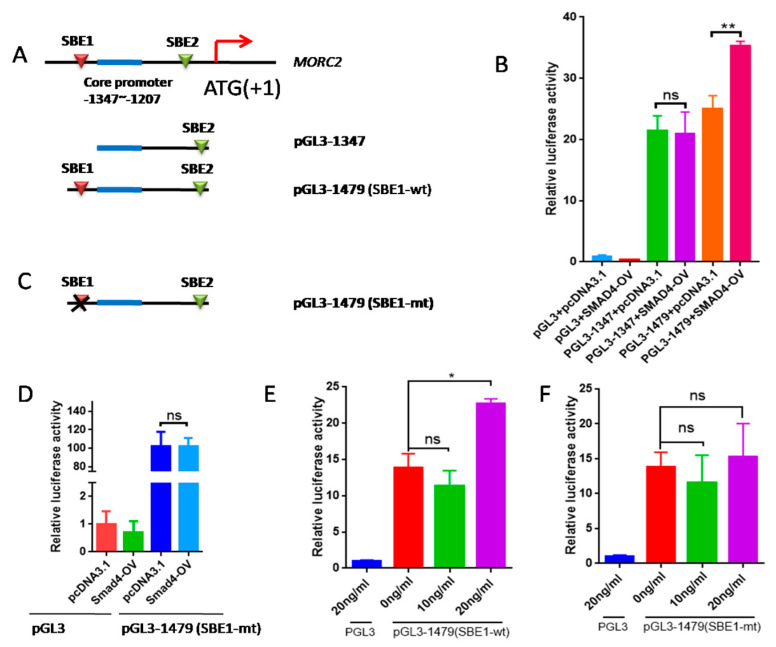
The transcription factor SMAD4 induces transcription of *MORC2*. (**A**) Schematic showing insertion of the *MORC2* promoter into the pGL3-basic vector. Potential SBE sites are indicated by red boxes. ATG was +1. (**B**) Changes of relative luciferase activity with or without SMAD4 over-expression. (**C**) The schematic of SBE1-mt in MORC2 promoter. mt indicates SBE site mutation. (**D**) Changes of relative luciferase activity of pGL3-1479 (SBE1-mt) with or without SMAD4 overexpression. (**E**,**F**) Changes in relative luciferase activity of the SEB1-wt or SBE1-mut MORC2 promoter treated with TGF-β1. Data are shown as mean ± SEM. (n = 3). * *p* < 0.05; ** *p* < 0.01; ns, no significance.

**Figure 7 ijms-23-10657-f007:**
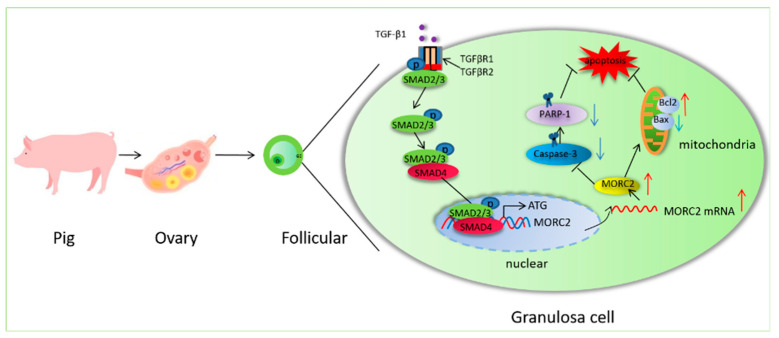
Schematic showing how the TGF-β/SMAD signaling pathway regulates porcine granulosa cell apoptosis by targeting MORC2. O:Oocyte; GC: granulosa cell. Red arrows indicate provementand the blue arrows indicate inhibition.

**Table 1 ijms-23-10657-t001:** The siRNAs used in this study.

siRNA	Sequence(5′ to 3′)
NC-siRNA	F: UUCUCCGAACGUGUCACGUTT R: UUCUCCGAACGUGUCACGUTT
SMAD4-siRNA	F: CACCAGGAAUUGAUCUCUCAGGAUU R: CACCAGGAAUUGAUCUCUCAGGAUU
MORC2-294	F: GCAAUACGGGAAUGGGUUATT R: UAACCCAUUCCCGUAUUGCTT
MORC2-584	F: GCGGAACGCUGGUCAUCAUTT R: AUGAUGACCAGCGUUCCGCTT
MORC2-1164	F: CCGCCUGAUCAAGAUGUAUTT R: AUACAUCUUGAUCAGGCGGTT

**Table 2 ijms-23-10657-t002:** Primers used for qRT-PCR and plasmids construction.

Name	Primer Sequence	Anneal T (°C)	Product Size (bp)	Usage
MORC2	F: TCACCAAGAAGGAAGACACTA R: AGATGATGACCAGCGTTCC	58.9	252	qRT-PCR
Bax	F: GCCGAAATGTTTGCTGAC R: GCCGATCTCGAAGGAAGT	58	154	qRT-PCR
Bcl2	F: TCAGGGATGGGGTGAACT R: TCAGAGACAGCCAGGAGAAAT	60	240	qRT-PCR
GAPDH	F: GGACTCATGACCACGGTCCAT R: TCAGATCCACAACCGACACGT	60	220	qRT-PCR
SMAD4	F: ATTGGTGTTCCATTGCCTAC R: TGGTCACTAAGGCACCTGAC	58	250	qRT-PCR
ATR	F: TATCTGGGCTCCCTCCTC R: CTCCTTTCTGTATTCCTGTAGAACT	58.6	204	qRT-PCR
ATM	F: CTGTGAGATTGGCGTTAGT R: AACATGCAAACTTGGTGAT	59	154	qRT-PCR
P53	F: TGACTGTACCACCATCCACTAC R: AGGCACAAACACGCACCT	60	149	qRT-PCR
Rad51	F: GGTGGAGGTGAAGGAAAG R: CTGGGTCTGGTGGTCTGT	60	156	qRT-PCR
PARP-1	F: AGCACCTGTGACGGGCTAC R: CTTGCTGATGTGCGAAGC	61	171	qRT-PCR
Caspase-3	F: TTGGACTGTGGGATTGAGACG R: CGCTGCACAAAGTGACTGGA	59	165	qRT-PCR
pcDNA3.1-his-v5-MORC2	F: CCCAAGCTTGCCACCATGGCTTTCACAAATTACAGCAGTCTGAATC R: CCGCTCGAGGTCCCCCTTGGTGATCAGGTCC	68	3102	Over-expression
pcDNA3.1-SMAD4	F: CCCAAGCTTGCCACCATGGACAATATGTCTATTACAAATACACCAACAAG R: CCGCTCGAGTCAGTCTAAAGGCTGTGGGTCGGC	68	1659	Over-expression
pGL3-MORC2-1347	F: CGGGTACCGGTCAAGAAAGCAGGAAGAGAG	68	1347	promoter
pGL3-MORC2-1479	F: CGGGTACCGCTTCTTCGAGTTAGGGGCCTGACT	68	1479	promoter
pGL3-MORC2-R	R: CCGCTCGAGTGCAGTGAGGTCTCCAGTTCTTT		-	promoter
pGL3-SBE1-mut	F: TGACTCCCCGGCGGATGGCTCCTCCGGTCCGC R: CATCTCGCCGGGGATCAGGCCCCTAACTCGAA			SEB mutation

## Data Availability

The original data in the article can be obtained directly from the corresponding author.

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
