# Peer review of "The TGF-β/SMAD Signaling Pathway Prevents Follicular Atresia by Upregulating MORC2"

_ijms, 2022, doi:10.3390/ijms231810657_

Round 1
Reviewer 1 Report
The article by Liu J et al is well written and organized. The methods used in the experimental part are modern and the research was well conducted. The authors should take care of several typos throughout the manuscript. If the Editors agree that the topic of the manuscript is suitable for publication in IJMS, I recommend this article for publication.
Author Response
Many thanks for the reviewer's recognition and encouragement, relevant exploration and research will continue. We carefully revised the manuscript as reviewer suggested.
Reviewer 2 Report
In the present manuscript, the authors examined the effect of MORC2 on porcine granulosa cell apoptosis and the role in the process of porcine follicular atresia regulated by the TGF-β/SMAD4 signalling pathway. The results provided are pretty new and expand the current knowledge about the mechanism underlying porcine follicular atresia, which is still quite limited. Such data, as presented above, can be primary for further research.
Generally, the methods and approaches used are property selected and designed. The results are interesting and clearly described. I pinpointed a few crucial details which should be corrected and clarified due to their significant impact on results reliability.
- First, the manuscript requires a solid editorial correction in terms of punctuation. There are a lot of punctuation errors (like no spaces, double spaces, or the size of the font used, e.g. lines 41, 56, 59, 65, 74, 77, 96, 105, 110,121, 137,139, 143-149…).
- Please provide information on the number of animals and cell cultures used in the experiment.
- Generally, the gene name is written in italics; please recheck the entire manuscript as it requires some revisions in this regard.
- Moreover, figures are difficult to read and should be replaced with bigger ones. Image G caption of Figure 1 is missing. Please also add a legend to the figures, as the used abbreviations (e.g. MORC-siRNA 294, MORC-siRNA 584, MORC-siRNA 1164, pcDNA-3.1-His-V5, and so on) are not understandable to everyone.
Other issues:
- Lines 52-54, 217-218, 239-241: These sentences need rewording
- Line 131: wrong reference
- Lines 157, 220, 287: “SBE”, “co-SMAD”, “CDS” - the first use of an abbreviation requires its full description.
- References 25 and 42 have not been cited in the text.
Author Response
In the present manuscript, the authors examined the effect of MORC2 on porcine granulosa cell apoptosis and the role in the process of porcine follicular atresia regulated by the TGF-β/SMAD4 signalling pathway. The results provided are pretty new and expand the current knowledge about the mechanism underlying porcine follicular atresia, which is still quite limited. Such data, as presented above, can be primary for further research.
Generally, the methods and approaches used are property selected and designed. The results are interesting and clearly described. I pinpointed a few crucial details which should be corrected and clarified due to their significant impact on results reliability.
- First, the manuscript requires a solid editorial correction in terms of punctuation. There are a lot of punctuation errors (like no spaces, double spaces, or the size of the font used, e.g. lines 41, 56, 59, 65, 74, 77, 96, 105, 110,121, 137,139, 143-149…).
Response: Thank you very much for your valuable comments. We apologize for the trouble this issue has caused you. That due to the template format, now we have solved this problem through communication with the edit office.
- Please provide information on the number of animals and cell cultures used in the experiment.
Response: Thanks. We have provide the information.
- Generally, the gene name is written in italics; please recheck the entire manuscript as it requires some revisions in this regard.
Response: Thanks very much. We have checked and revised them.
- Moreover, figures are difficult to read and should be replaced with bigger ones. Image G caption of Figure 1 is missing. Please also add a legend to the figures, as the used abbreviations (e.g. MORC-siRNA 294, MORC-siRNA 584, MORC-siRNA 1164, pcDNA-3.1-His-V5, and so on) are not understandable to everyone.
Response: Thanks very much. We have added it.
Other issues:
- Lines 52-54, 217-218, 239-241: These sentences need rewording.
Response: Thank you very much for your valuable advice. We have reworded them.
- Line 131: wrong reference
Response: Thanks very much. We have revised it.
- Lines 157, 220, 287: “SBE”, “co-SMAD”, “CDS” - the first use of an abbreviation requires its full description.
Response: Thanks. The relative abbreviation have been fully description.
- References 25 and 42 have not been cited in the text.
Response: Thanks very much. We have revised it.